# The Spatial Relationship and Evolution of World Cultural Heritage Sites and Neighbouring Towns

Yihan Xie [1,2,3,4], Ruixia Yang [1,2,4,*], Yongqi Liang [1,2,3,4], Wei Li [1,2,3,4] and Fulong Chen [1,2,4]

1   Key Laboratory of Digital Earth Science, Aerospace Information Research Institute, Chinese Academy of Sciences, Beijing 100094, China
2   International Research Center of Big Data for Sustainable Development Goals, Beijing 100094, China
3   University of Chinese Academy of Sciences, Beijing 100049, China
4   International Centre on Space Technologies for Natural and Cultural Heritage under the Auspices of UNESCO, Beijing 100094, China
*   Correspondence: yangrx@aircas.ac.cn

**Abstract:** The past few decades have witnessed unprecedented global urbanisation, with direct or indirect impacts on global cultural heritage sites. Research on the spatial relationship between cultural heritage sites and urban areas has provided a new perspective for understanding the impact processes between them, which have previously been discussed at the regional scale. In this article, we analyse the spatial relationship between world cultural heritage sites and neighbouring towns through systematic observations at the global scale and attempt to model change processes and identify impact mechanisms. We adopt spatial analysis and spatial statistics to analyse the changing characteristics of the spatial relationship between world cultural heritage sites and neighbouring towns from 1990 to 2018 and to analyse the impact processes at different spatial and temporal scales by combining indicators, such as income levels and urbanisation rates, at the national scale. The results show that 8.52% of world cultural heritage sites have been incorporated into urban areas over the aforementioned 28 years, with a certain aggregation in the spatial distribution of these sites, and that the growth rate can be divided into three phases, including two periods of rapid growth. The spatial relationship between towns and the 523 world cultural heritage sites that were previously located outside towns has not yet changed substantially, but the distances between most of the towns and these sites have been decreasing, with 81% of the world cultural heritage sites having a variation in distance from the corresponding town of 7.60 km or less. We also analysed the variation in distance between cultural heritage sites and neighbouring towns and found a relationship with indicators, such as the income level and urbanisation rate of the country to which a site belongs. Among the indicators, variation in national urbanisation rates most greatly affected the distance between heritage sites and towns. This study shows that world cultural heritage sites are affected by urbanisation and that particular attention should be given to the relationship between cultural heritage sites and neighbouring towns, especially in countries undergoing rapid urbanisation, so that the authenticity and integrity of cultural heritage are not compromised. This article provides a basis for development plans and policies in urban design, especially those that are sensitive to cultural heritage, and may also provide ideas and references for heritage conservation against the background of urbanisation.

**Keywords:** world cultural heritage; urban areas; spatial relationship analysis; heritage conservation

## 1. Introduction

World cultural heritage sites are shared global treasures and economic, cultural and social drivers of sustainable development [1]. In 2015, the United Nations Sustainable Development Summit formally adopted the 2030 Agenda for Sustainable Development, which identifies 17 Sustainable Development Goals (SDGs) and 169 subgoals to be achieved over the next 15 years [2]. SDG 11, "Building inclusive, safe, resilient and sustainable cities

and human settlements", has seven subgoals, including SDG 11.4, "Strengthening efforts to protect and safeguard the world's cultural and natural heritage". In 2020, the International Council on Monuments and Sites (ICOMOS) published its 2020 The ICOMOS World Report 2016–2019 on Monuments and Sites in Danger (Heritage at Risk) [3]. Numerous publications over the last 20 years have showcased many problematic and difficult heritage sites, thereby drawing the attention of a wider audience to the risks posed by various destructive human activities and economic overdevelopment.

Echoing the UN SDGs and the heritage conservation strategy of ICOMOS, this article takes the perspective of urban sprawl to offer new ideas for monitoring the state of listed heritage sites in different global development contexts. Recent decades have witnessed unprecedented rapid urban expansion across the globe; this expansion has directly or indirectly affected world cultural heritage sites. As this issue is gradually gaining academic attention, related research has also shown an increasing trend. For example, in 2015, Athos Agapiou et al. used ETM+ and TM satellite images to study and analyse the spatial patterns of urban sprawl to monitor urbanisation and the growth dynamics of urbanisation in the Paphos region of southwestern Cyprus during recent decades and to assess the impact on monuments and archaeological sites [4]. In 2018, Gwendolyn Kristy et al. used historical aerial and satellite imagery, historical maps and gazetteers, modern urban planning data and predictive models in a geographic information system (GIS) to track recent urban growth, calculate potential areas of future urban growth and investigate patterns of urbanisation impacts on cultural heritage in Herat [5]. Additionally, in 2018, Yingying Zhang et al. used night-time lighting data and other methods to assess urbanisation variations in 152 Buddhist monasteries on the Tibetan Plateau between 1993 and 2013 [6]. In 2020, Da Xiao et al. developed an urbanization intensity index (UII) to quantitatively measure urban dynamics in the vicinity of World Heritage sites in the Belt and Road region [7]. Previous studies have some common findings that are of reference value to this study; these findings include the view that urban sprawl puts enormous pressure on cultural heritage sites [4–13] and needs to be brought to the relevant authorities' attention, especially regarding improved monitoring, legislation/enforcement, awareness raising and education, and enhanced coordination [7]. However, previous studies have focused mainly on a single region, without systematic observation and macroscopic knowledge on a global scale, or have focused on cultural heritage sites already present in urban areas but have rarely monitored or assessed cultural heritage sites outside urban areas, and the findings of these studies are sensitive to distance (from urban areas) on a large scale [4–13].

In view of this gap in the literature, this article selects the "global" scale as the study area to explore the spatial relationship between world cultural heritage sites and neighbouring towns to demonstrate the processes of variation and to analyse the mechanisms of influence. This article provides a global case study for using big data to support cultural heritage conservation, provides a basis for development plans and policies that are sensitive to cultural heritage in urban planning, and provides an idea and reference for heritage conservation in the context of urbanisation, thereby echoing the UN SDGs and ICOMOS heritage conservation strategies and supporting the global conservation and sustainable development of world heritage.

In the remainder of this article, we introduce the data requirements (Section 2) and the method framework (Section 3), report the results (Section 4) and discuss some influencing factors and the anomalies in the results (Section 5). Finally, we draw some conclusions from this study (Section 6).

## 2. Data Collection and Collation

In general, we need to collect heritage ontology data and global urban boundary data to analyse the characteristics and evolution of the spatial relationships between cultural heritage sites and their neighbouring towns. In addition, we need national-scale socioeconomic data on the country to which the heritage belongs to analyse the influencing

mechanisms for distance variations between these sites and neighbouring towns. Therefore, the data requirements for this article are as follows.

### 2.1. Attribute Data

The attribute data used in this study are world cultural heritage data and national-scale socioeconomic data: (1) information on 869 world cultural heritage sites; as of 2019, this information is sourced from the Global World Heritage dataset [14] (with data obtained from relevant official websites, open-source encyclopaedias, companies, universities, research institutes and organisations) (Table 1), with selected data items, such as heritage type, location, and country and region to which the sites belong; (2) World Bank–sourced data on income levels, land area, population size and urbanisation rates by country for 1990–2018; (3) heritage-associated city attributes obtained from the Simple Maps website, which contains the locations and populations of 15,493 cities in 224 countries and regions.

**Table 1.** Data source and coverage introduction list.

| Content | Source | Time Scale | Number of Data |
|---------|--------|------------|----------------|
| Standard heritage data | Global World Heritage dataset [14] | Up to 2019 | 869 |
| Heritage ontology data | Global World Heritage dataset [14] | Up to 2019 | 869 |
| Gross national income (GNI) per capita | World Bank | 2018 | 173 |
| Area (national scale) | World Bank | 2018 | 212 |
| Population (national scale) | World Bank | 1990–2018 | 60 |
| Urbanisation rate (national scale) | World Bank | 1990–2018 | 70 |
| City location and population | Simple Maps company | Up to 2019 | 15,493 |

National-scale socioeconomic patterns influence a higher dimension of heritage management and conservation policies. To explain this issue in more detail, four indicators for the countries in which the heritage sites are located (Table 1) were selected for further study: income level, land area, population size and urbanisation rate. In the case of world cultural heritage sites, which may be located in multiple countries, the indicators are linked to site attribute data by country name.

### 2.2. Global Urban Boundary (GUB) Dataset

Artificial impervious areas are mainly man-made structures that are composed of any material that impedes or prevents natural infiltration of water into the soil. They include roofs, paved surfaces, hardened grounds, and major road surfaces mainly found in human settlements [15]. The change of global artificial impervious area (GAIA) is a critical indicator for understanding the impact of global urbanization on human society and the environment [15]. In this article, we apply it to the monitoring and conservation of cultural heritage innovatively.

Many scholars have already investigated the extraction of impermeable surfaces. Early impervious surface extraction mainly included ground measurement and manual digitisation methods [16]. Although these methods are highly accurate, their application is limited by the low degree of automation, time and effort, and the limited range of data obtained. As early as ten years ago, the development of remote sensing technology and remote sensing image processing technology made it possible to obtain a large range of impervious surfaces quickly and at a relatively low cost, which is of increasing interest to various fields [17]. Until the present moment, the remote sensing method has become the most efficient and necessary way to obtain urban artificial impervious area data on a global scale, continuous over time and with relatively consistent accuracy. Therefore, remote sensing datasets are essential and fundamental data for this study.

We have studied and analysed a number of global urban remote sensing data products such as 2000–2020 Global Urban Land Use/cover Composites with 30 m spatial resolution (GULUC-30) from Wenhui Kuang's team [18], 1990–2010 Multi-temporal urban land products and reference datasets by calculating Normalized Urban Areas Composite Index (NUACI) from Xiaoping Liu's team [19], 1975–2015 The Global Human Settlement Layer

(GHSL) supported by the Joint Research Centre (JRC) [20], 1990–2018 Annual maps of global artificial impervious area (GAIA) from Gong Peng's team [15]. Taking into account the characteristics of each dataset, the GAIA dataset was finally selected in terms of its appropriate spatial and temporal resolution.

The multitemporal (1990, 1995, 2000, 2005, 2010, 2015 and 2018) GUB dataset was developed by Prof. Gong Peng's team at Tsinghua University in 2020 based on the developed global high-resolution artificial impervious surface mapping product (Global Artificial Impervious Area (GAIA)) [15]. The dataset contains data on 9110 towns and has a spatial resolution of 30 m and a period of 1990–2018 (Table 2).

**Table 2.** Characteristics of the Global Urban Boundary (GUB) dataset.

| Year | Number of Towns | Minimum Area (km$^2$) | Maximum Area (km$^2$) | Average Area (km$^2$) |
|------|------|------|------|------|
| 1990 | 6470 | 1 | 5861 | 567 |
| 1995 | 7294 | 1 | 6200 | 662 |
| 2000 | 7710 | 1 | 6595 | 702 |
| 2005 | 8247 | 1 | 6874 | 762 |
| 2010 | 8444 | 1 | 7755 | 801 |
| 2015 | 8561 | 0.75 | 8294 | 818 |
| 2018 | 9110 | 1 | 10,604 | 897 |

This dataset was applied to the analysis of distance variation between global heritage sites and urban areas in the article, in addition to the demonstration of distance variation processes in case heritage sites.

## 3. Methods

In this article, we take 869 global World Heritage Sites (as of 2019) and the surrounding towns as the research objects and choose spatial analysis and spatial statistics as the key methods. We analyse the changing characteristics of the spatial relationship between global world cultural heritage sites and the surrounding towns from 1990 to 2018 and reproduce the change processes of these places. Eventually, we integrate indicators, such as income levels and urbanisation rates, at national scales to analyse the influencing mechanisms at different spatial and temporal scales (Figure 1).

### 3.1. Spatial Relationship Patterns

The spatial relationship between cultural heritage sites and towns is abstracted as a spatial relationship between two planar polygons. In GIS topology theory, there are eight spatial relationships between two polygons; these relationships cover all cases and are mutually exclusive: disjoint, touch, overlap, equal, contain, inside, cover and coverby (Figure 2).

Based on the possible forms of spatial relationships between cultural heritage sites and towns, the above eight spatial relationships have been simplified into three, namely, disjoint, overlap and contain, from which the concepts of spatial relationships in the following are derived (Figure 3).

This method of concept abstraction facilitates our judgement and analysis of the spatial relationships between World Heritage Sites and towns in ArcGIS.

### 3.2. Gini Coefficient Analysis

Cultural heritage is a product of human activity, and the distribution of cultural heritage is not random. We chose the Gini coefficient, which portrays the distribution of spatial elements in discrete regions, to determine the spatial distribution characteristics of various types of world cultural heritage sites on a global scale by using the following formula:

$$\text{Gini} = \frac{-\sum_{i=1}^{X} S_i * \ln S_i}{\ln X} \tag{1}$$

where $S_i$ is the number of world cultural heritage sites in the *i*th region as a proportion of the total number of world cultural heritage sites worldwide and $X$ is the number of regions analysed; in the text, $X$ is five. Theoretically, the Gini coefficient lies between 0 and 1, with larger values indicating greater concentration.

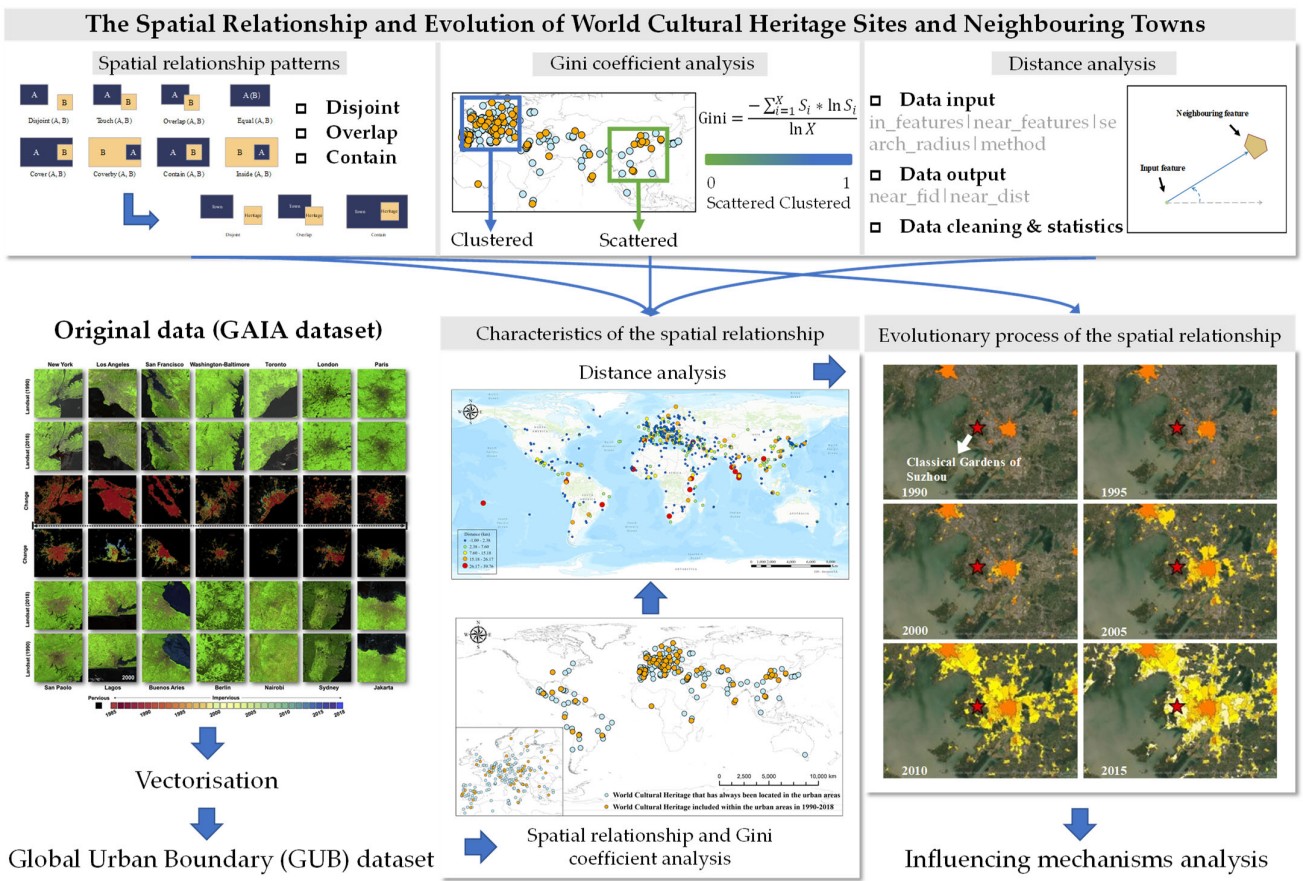

**Figure 1.** Flow diagram of the methods used in this study.

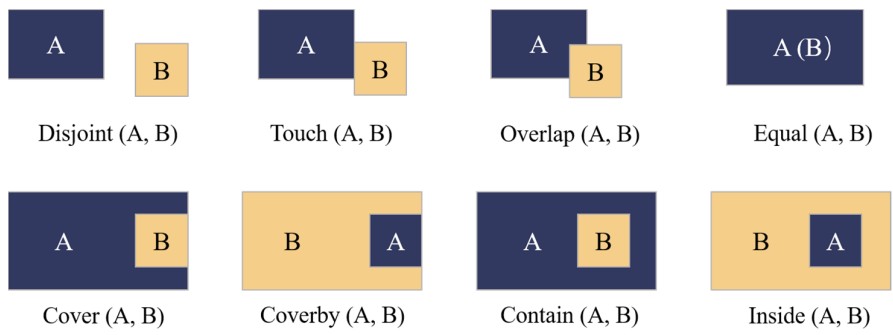

**Figure 2.** Eight spatial relations in GIS topology theory.

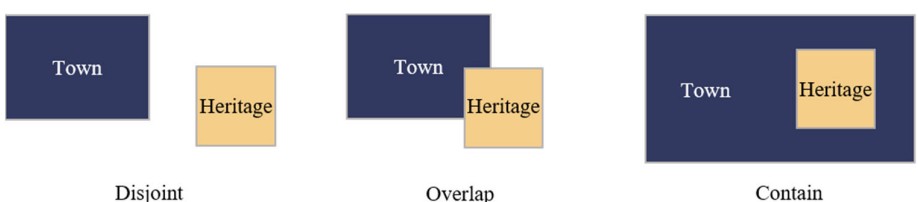

**Figure 3.** Simplified spatial relationships between cultural heritage sites and towns.

*3.3. Distance Analysis*

The data used for distance analysis are heritage attribute data and the GUB dataset, which is the vectorial boundary of the artificial impervious surface mapping product (Global Artificial Impervious Area (GAIA)).

The GAIA dataset mapped urban built-up areas and artificial impervious areas using the full archive of 30 m resolution Landsat images on the Google Earth Engine platform. With ancillary datasets, including night-time light data and Sentinel-1 Synthetic Aperture Radar data, the mean overall accuracy is higher than 90%. The author of the dataset, Gong Peng et al., has previous studies that suggest that the mapping performance in arid and semi-arid regions is relatively poor due to the highly confusing spectral information between the artificial impervious area and the surrounding bare lands [21], resulting in an over-commission of artificial impervious areas in dryland regions. Therefore, in the research of GAIA dataset, they divided the whole world into arid areas and non-arid areas according to the global biome map [22]. For non-arid areas, they used a reliable impervious surface mapping algorithm [21]. In arid regions, they developed an improved algorithm by including additional constraints to mitigate the overestimation caused by spectral confusion between impervious areas and bare lands. To improve the mapping efficiency on the GEE, they divided the whole world into 583 geographical grids (3.5° × 3.5°), of which 155 grids contain arid biomes. The mapping procedures were implemented in all grids. Aided by the temporal contexts used in their algorithms, the missing of clear observations in extremely cloudy regions was mitigated considerably using temporally close Landsat images as well as all available clean pixels in highly cloudy Landsat scenes [15].

Considering the characteristics of the GAIA and GUB datasets comprehensively, we think they are a good representation of how urban areas around the world are changing. Therefore, we described the spatial relationship between the World Heritage Sites and the boundaries of the towns in the GUB dataset by calculating the distance between them.

Using the ArcGIS *near* analysis of *Proximity toolset* (illustrated in Figure 4), we calculated the distance between the input element and the nearest element in another layer or element class within a specified search radius. The method involves concise steps, rapid calculations and accurate results; the search radius is determined from the data, thus avoiding the problem of blindly determining the search range.

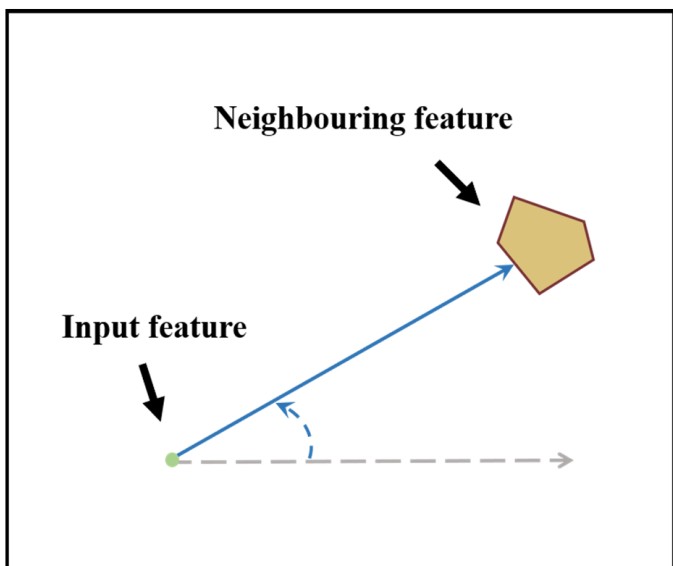

**Figure 4.** Calculation of the minimum distance between a heritage site and the neighbouring town boundary.

The calculation steps are as follows:

1. Data input

There are four parameters to be input, which are "in_features", "near_features", "search_radius" and "Method". Among them, the basis for setting the search radius was that experimentally, the farthest distance from a city boundary of all cultural heritage sites worldwide from 1990 to 2018 was 3918 km; thus, it is reasonable to choose a search radius of 4000 km. And the GEODESIC method was chosen because the it considers the curvature of the spheroid and correctly addresses data near the dateline and poles. The default method PLANAR, however, calculates only planar distances (Table 3).

**Table 3.** Input data form for the *near* tool.

| Parameter | Content |
|---|---|
| in_features | Heritage sites that have always been located outside town boundaries |
| near_features | Global Urban Boundary from 1990 to 2018 |
| search_radius | 4000 km |
| Method | GEODESIC (geodesic distances) |

2. Data output

The output parameters for the *near* tool are "NEAR_FID" and "NEAR_DIST". The "NEAR_FID" is the object ID of the nearest neighbouring element; if no neighbouring element is found, the value is −1. The "NEAR_DIST" is the distance between the input element and neighbouring elements. The value is in linear units of the input element's coordinate system, or "metres" when the method parameter is set to GEODESIC and the input is in a geographic coordinate system. If no neighbouring elements are found, the value is −1. Once the *near* tool has been run, the output item fields are automatically added to the world cultural heritage table.

3. Data cleaning and statistics

By observing and sampling the initial distance data and comparing them with the actual situation, we found that the distance calculation results had extreme values and unreasonable situations caused by the boundary extraction error. Therefore, we observed the changes in images of different years. After individual verification, the least-squares fitting of some original data was carried out, a few missing data were supplemented, and the extreme value was reasonably corrected.

## 4. Results

### 4.1. Analysis of the Characteristics and Evolution of the Spatial Relationship between Cultural Heritage Sites and Neighbouring Towns

4.1.1. Characteristics of the Spatial Relationship between Heritage Sites and Towns

Globally, 74 world cultural heritage sites (8.52% of the total, mainly industrial heritage sites, historic towns and archaeological sites) were incorporated into the neighbouring towns from 1990 to 2018; the remaining 795 heritage sites, of which 272 were in urban areas and 523 were located outside urban areas, did not change their spatial relationship with the neighbouring towns. The 74 heritage sites that were incorporated into neighbouring towns were located mainly in Europe and North America (47 heritage sites, 61.0%), thus indicating that rapid and significant urban expansion in developed regions has altered the natural and social environment in which heritage sites are located and that these sites require attention. Of the 523 heritage sites that remained outside urban areas, 235 (44.9%) were located in Europe and North America. In terms of the proportion of regions, more than 70% of the cultural heritage sites in less-developed regions, such as Africa and the Asia-Pacific region, were still located outside urban areas.

To quantitatively describe the degree of agglomeration of the 74 heritage sites that were incorporated into neighbouring towns and determine their overall spatial distribution

at the scale of the major regions of the world, we calculated the Gini coefficient. We divided the world into five regions ($X = 5$), and the specific parameters are shown in Table 4, where $S_i$ is the proportion of the number of world cultural heritage sites in the $i$th region to the total number of world cultural heritage sites worldwide.

**Table 4.** Value of the Gini coefficient parameter.

| Region | $i$ | $S_i$ |
|---|---|---|
| Europe and North America | 1 | 0.613 |
| Latin America and the Caribbean | 2 | 0.107 |
| Asia-Pacific | 3 | 0.200 |
| Arab region | 4 | 0.053 |
| Africa | 5 | 0.027 |

The results of the calculation show that the Gini coefficient for the 74 heritage sites that were incorporated into towns is 0.6918. Theoretically, the closer the value of the Gini coefficient is to 1, the higher the concentration; therefore, the 74 heritage sites that were incorporated into towns are spatially clustered and concentrated in the European region (Figure 5).

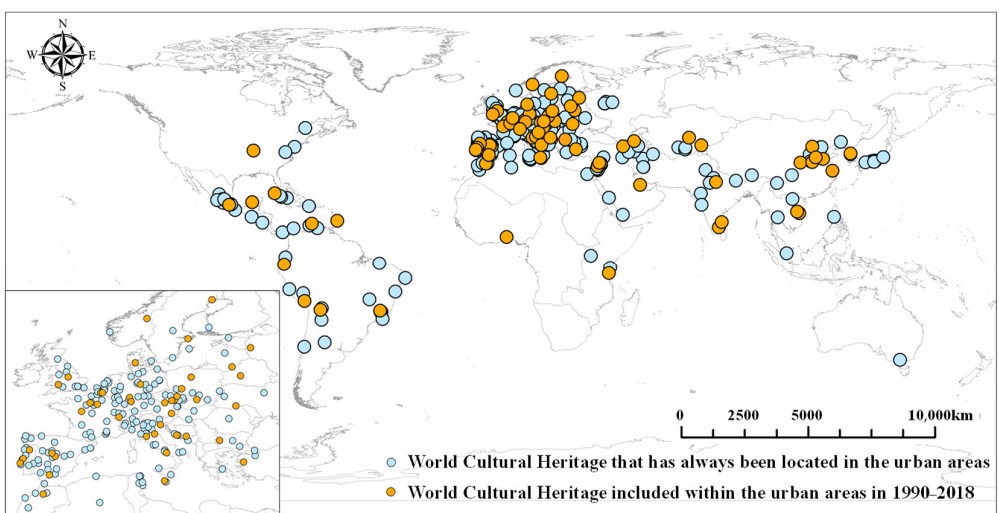

**Figure 5.** Spatial distribution of world cultural heritage sites.

Of the 523 world cultural heritage sites that were located outside towns from 1990 to 2018, only three experienced an increase in the distance between them and the towns, and all of those distances changed by less than 1 km; 33 experienced no variation in distance from the towns; and the distances between the remaining heritage sites and towns decreased to varying degrees, with the largest decrease being 39.76 km. According to the statistics, 81% of the world cultural heritage sites had a variation in distance from the town of 7.60 km or less, and the number of heritage sites with a significant distance variation was small, as shown in Figure 6a.

According to the distance analysis of world heritage sites and the boundaries of the towns in the GUB dataset, from 1990 to 2018, the average distance between world cultural heritage sites and neighbouring towns showed mostly declined. However, although not significant, the average distance increased slightly in 2015–2018, as shown in Table 5 and Figure 6b. This finding indicates that some countries and regions may have started to pay attention to the issue of the impact of urban areas on cultural heritage sites and to take corresponding measures in planning towns and cities.

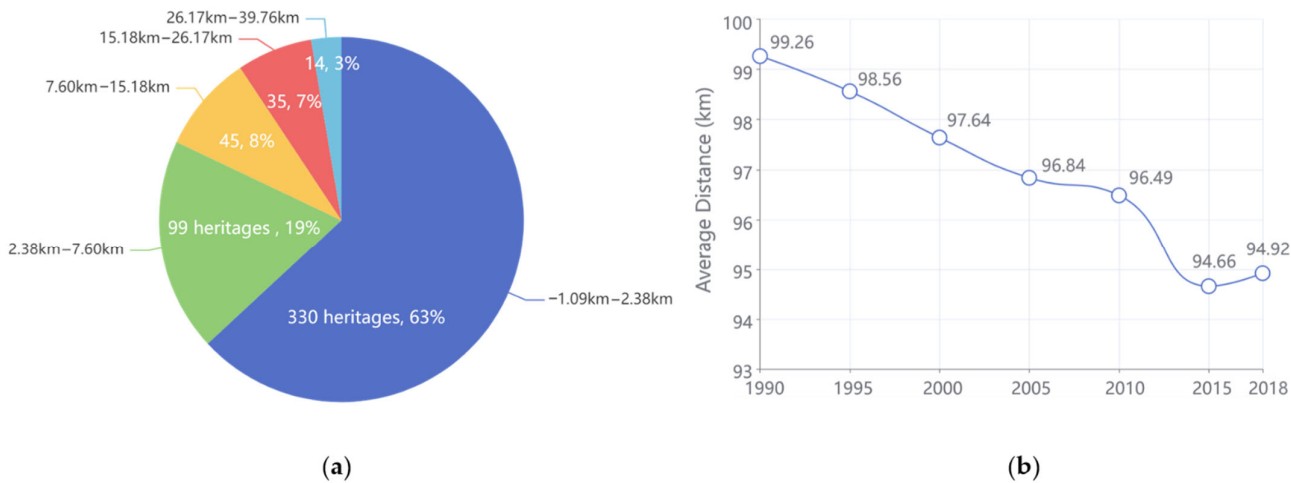

**Figure 6.** Distance variation between world cultural heritage sites and neighbouring towns in 1990–2018: (**a**) distribution of distance variation; (**b**) the trend of variation in average distance.

**Table 5.** Average rate of distance variation between world cultural heritage sites and neighbouring towns.

| Periods | Rate of Distance Variation (km/year) | Description |
| --- | --- | --- |
| 1990–2005 | 0.14–0.18 | Moderate |
| 2005–2010 | 0.07 | Quite slow |
| 2010–2015 | 0.37 | Fast |
| 2015–2018 | −0.09 | Small increase in distance |

Further statistics on the distribution of world cultural heritage sites with a variation greater than 7.60 km (Figures 7 and 8) show that these sites are located mainly in Europe and North America (39.36%) and in the Asia-Pacific region (37.23%); a few sites are also located in Latin America and the Caribbean, Africa, and the Arab region, and these sites are located mainly in coastal areas. This finding suggests that the variations in distances from towns may be related to the development of the country or region in which the cultural heritage site is located.

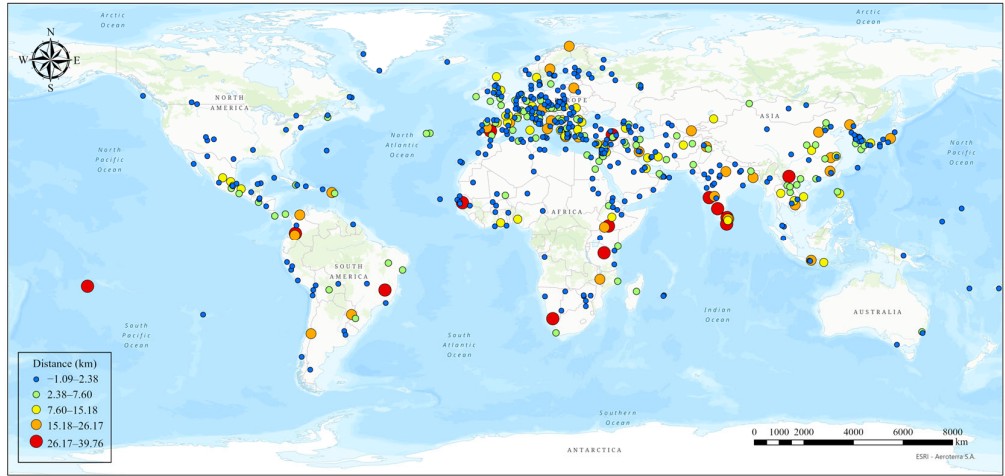

**Figure 7.** Distance variation between world cultural heritage sites and neighbouring towns in 1990–2018.

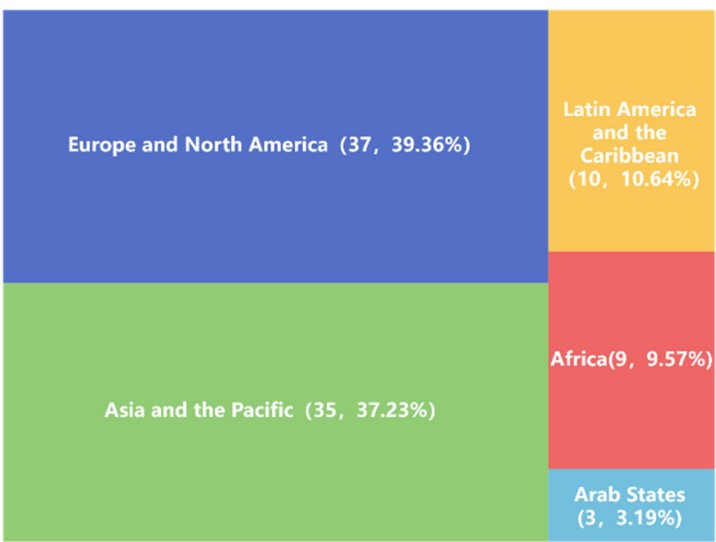

**Figure 8.** Distribution of world cultural heritage sites with a variation in distance greater than 7.60 km from neighbouring towns in 1990–2018.

### 4.1.2. Analysis of the Evolutionary Process of the Relationship between Heritage and Towns

The process of change in the relationship between the 869 world cultural heritage sites and neighbouring towns was analysed and mapped (Figure 9). The results indicate that cultural heritage sites were incorporated into urban areas in all periods from 1990 to 2018.

A count of world cultural heritage sites that have been incorporated into neighbouring towns since 1990 (Figure 10) shows that the number of these sites has been increasing, and this increase can be divided into three phases according to the rate of increase. Of the three phases, the first phase represents the most significant increase, with 37 heritage sites representing 50% of the total number of sites incorporated into neighbouring towns during the 28-year period; the third phase also represents a high increase, with 16 heritage sites incorporated into neighbouring towns in just three years; this increase represents a peak of 21.62% of the total number of sites incorporated into neighbouring towns during the 28-year period.

In response to the above results, we believe that in regard to the global scale, the reasons are multiple and most likely related to the fluctuating level of average economic development and the global urbanisation rate. However, as an important phenomenon, the impact of urban expansion on cultural heritage sites has attracted sustained attention in recent years.

Then, we counted the cultural heritage sites that were incorporated into urban areas by region (Figure 11). The findings indicate that most of these sites were incorporated in Europe and the Asia-Pacific region, while elsewhere, sites were incorporated only between 1990 and 2005.

For this article, we selected four case heritage sites: the Classical Gardens of Suzhou in China, the Pampulha Modern Ensemble in Brazil, the Major Mining Sites of Wallonia in Belgium and the Dilmun Burial Mounds in Bahrain, and we extracted the extent of their proximity to urban areas in different periods by using high-definition remote sensing images to visually demonstrate the expansion of towns around the four heritage sites from 1990 to 2018 (Figure 12). The spatial relationship between the four heritage sites and the neighbouring towns followed a similar trend, with the heritage sites being adjacent to the towns or still at a distance from the towns in 1990 and then as the town areas continued to expand, being completely incorporated into the towns by 2018.

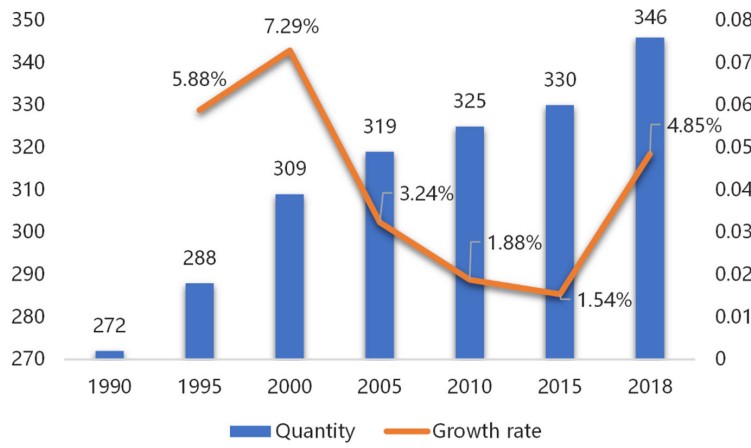

**Figure 9.** The evolution of the spatial relationship between world cultural heritage sites and neighbouring towns from 1990 to 2018.

**Figure 10.** Incorporation of world cultural heritage sites into urban areas from 1990–2018.

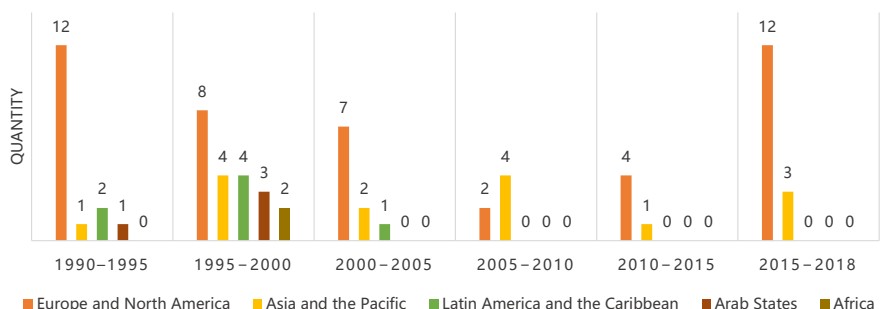

**Figure 11.** Regional statistics for world cultural heritage sites incorporated into towns in 1990–2018.

**Figure 12.** The process by which cultural heritage sites are incorporated into towns.

*4.2. Influencing Mechanisms for Distance Variations between World Cultural Heritage Sites and Neighbouring Towns*

4.2.1. The Impact of Urban Expansion

Positive Impact

1.  Contribution to cultural transmission

When cultural heritage is valued and protected by the city, the city will provide an excellent platform for presentation and promotion. By collecting, collating and publishing documentation and promoting various exhibitions, tours and exchange meetings, the city will enable an increasing number of people to gain a deeper knowledge of cultural heritage and to participate in its preservation, transmission and promotion (Figure 13).

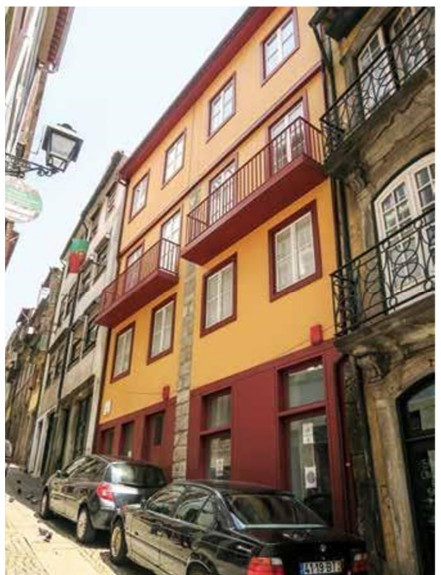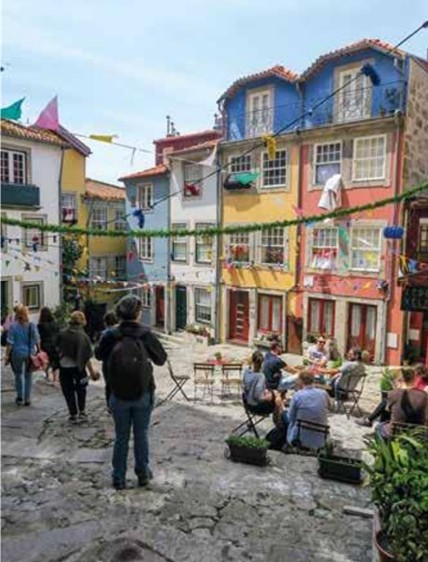

**Figure 13.** In Porto, old houses that were renovated and offered as holiday flats to visitors [3].

2.  Contribution to the development of the local economy and tourism

For developing regions and countries, there is a complex link between urban poverty and cultural factors. If cultural elements (cultural heritage, etc.) are made more marketable, they may generate satisfactory economic returns [23]. For any region, cultural heritage preservation has also been shown to benefit urban and regional development [24], especially tourism and economic development, which will receive a direct boost. Figure 14 shows people visiting and shopping on Tunxi Old Street (a cultural heritage site in Huangshan City, China) during the Spring Festival.

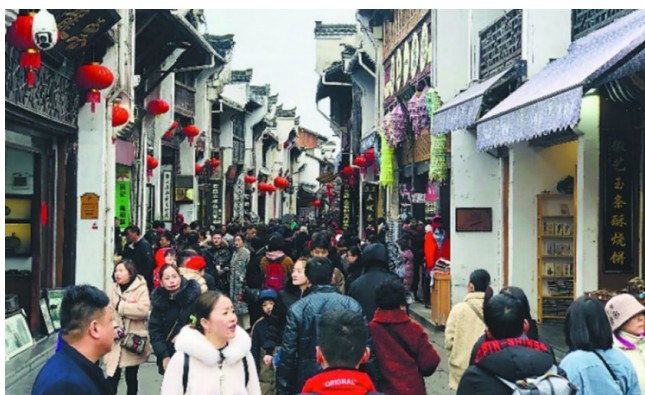

**Figure 14.** Cultural heritage promotes tourism and the economic development of the town [25].

Negative Impact

3. Direct destruction of cultural heritage by urban expansion

During urbanisation, the demolition and renovation of dilapidated houses, real estate development and infrastructure construction are often inevitable [26]. Some people disregard cultural heritage sites and subjectively demolish them without permission in pursuit of economic benefits or development space [27,28]; demolishing these sites can greatly damage cultural heritage (Figure 15). Even if not demolished, cultural heritage sites are exposed to continuous vibrations triggered by infrastructure construction [4], the indirect effects of pollutants emitted by heavy vehicles and machinery [4], and unconscious human damage during the transformation of the surrounding environment.

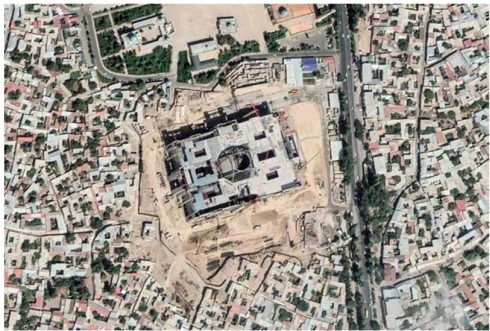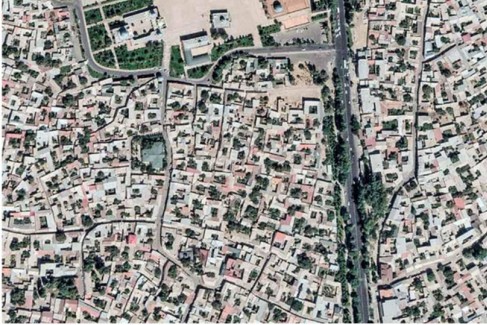

**Figure 15.** The imamate of Hazrat complexes disappeared during the expansion of the town [3].

4. Restoration destroys the original historical character of the cultural heritage

During urbanisation, not all cultural heritage sites are destroyed, but much renovation work is conducted, such as relocation [29], reconstruction [30,31], continued use [32], transformation into museums [33,34] (Figure 16), development of tourist sites [35,36], or several of these phenomena at the same time. In many cases, the principles of "restoration to original condition" are seriously violated, resulting in the destruction of cultural heritage sites through excessive maintenance and irreparable damage [26].

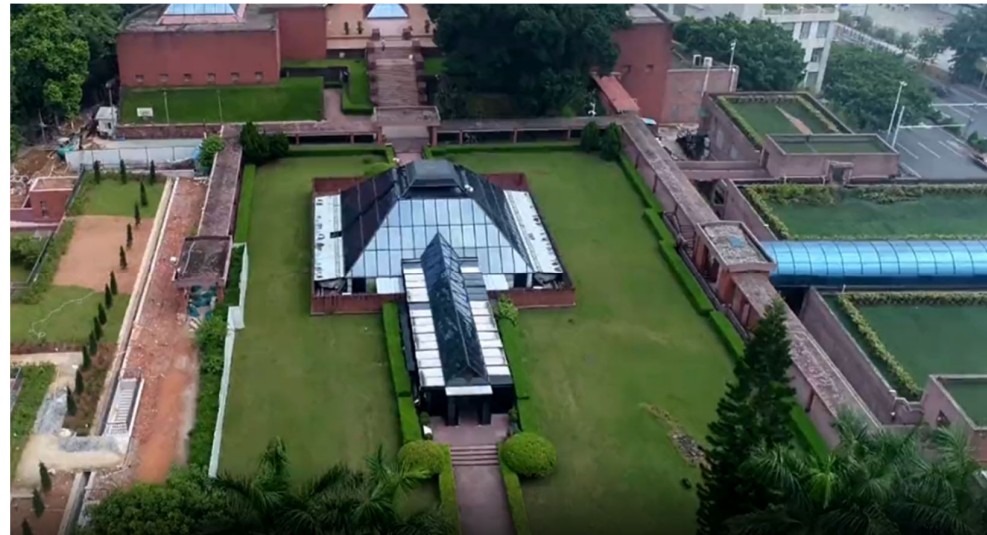

**Figure 16.** The tomb of the Western Han Dynasty in the city has been converted into a museum [37].

5. Destruction of the overall historical and cultural environment in urban development and construction

During urban construction, a holistic mindset is often lacking, and cultural connotations are often rarely considered in urban design. Not considering cultural connotations

can lead to the destruction of the environment around cultural heritage sites, in marked contrast to the overall style of the city [26] (Figure 17).

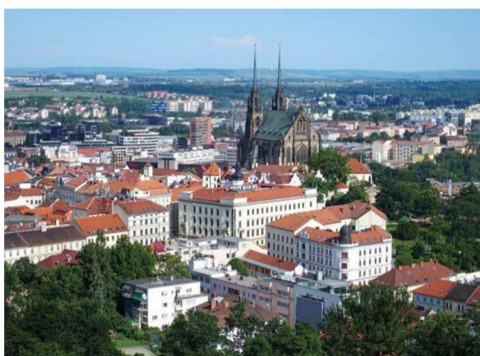 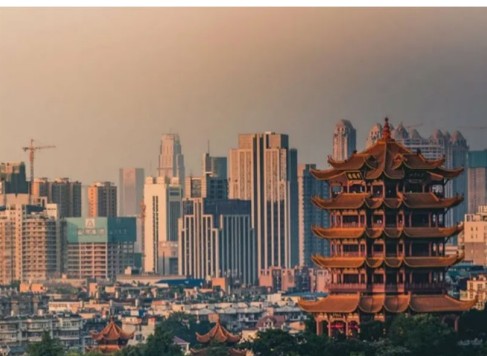

**Figure 17.** Historical and cultural heritage buildings in modern urban areas (Notre-Dame de Paris in France [3] and Yellow Crane Tower in China [38]).

Therefore, to conserve cultural heritage, planners must focus on urbanisation. Today, little cultural heritage is left to preserve, and we should hasten to salvage and include more precious cultural heritage in our conservation efforts, both for the present and for future generations [39].

4.2.2. Analysis of Factors Influencing Distance Variations between Heritage Sites and Neighbouring Towns

Considering the social and economic factors that influence the relationship between heritage sites and neighbouring towns, we selected national-scale indicators, namely, income level, land area, population and urbanisation rate, to analyse the influence mechanisms of variations in distance between heritage sites and neighbouring towns. We graded the 60 countries by using complete data according to official statistics and the Jenkspy grading method (Table 6). This method grades (classifies) the data into groups ranging from large to small, minimises intraclass differences and maximises interclass differences. The variation in population and variation in the urbanisation rate were calculated as follows.

$$P = P_{2018} - P_{1990} \tag{2}$$

$$U = U_{2018} - U_{1990} \tag{3}$$

where P is the variation in population of a country from 1990 to 2018, $P_{2018}$ is the population of a country in 2018, $P_{1990}$ is the population of a country in 1990, U is the variation in the urbanisation rate of a country from 1990 to 2018, $U_{2018}$ is the urbanisation rate of a country in 2018, and $U_{1990}$ is the urbanisation rate of a country in 1990.

In terms of the impact factor indicators (Figure 18), the distance variation between cultural heritage sites and neighbouring towns is generally higher in countries with high and medium-high income levels, as these countries have more developed economies, grow faster and experience greater urban expansion. The relationship between distance variation and the land area of the country in which a site is located is normally distributed, i.e., cultural heritage sites with large distance variations are found not in countries with very large or very small land areas but in medium-sized countries. Countries with moderate and large population variations also have an overall greater distance variation, as increases in population require more land for construction; this finding is reflected in a greater distance variation between world cultural heritage sites and neighbouring towns. Finally, the distance variation between cultural heritage sites and neighbouring towns is most pronounced in countries with greater variations in urbanisation rates, thus further suggesting that urbanisation and urban expansion processes are directly responsible for distance variations between cultural heritage sites and neighbouring towns.

**Table 6.** Sixty countries are graded for each indicator.

| Gross National Income (GNI) per Capita in 2018 (Current Price in USD) | | | |
|---|---|---|---|
| **Grade** | **GNI** | **Number of countries** | **Description** |
| 1 | <2010 | 4 | Low income |
| 2 | 2010–3820 | 10 | Low–medium income |
| 3 | 3820–8070 | 14 | Medium income |
| 4 | 8070–14,150 | 8 | Medium–high income |
| 5 | >14,150 | 24 | High income |

| Land area (km$^2$) | | | |
|---|---|---|---|
| **Grade** | **Land area** | **Number of countries** | **Description** |
| 1 | $<2.46 \times 10^5$ | 24 | Small land area |
| 2 | $2.46 \times 10^5$–$7.89 \times 10^5$ | 18 | Small–medium land area |
| 3 | $7.89 \times 10^5$–$1.68 \times 10^6$ | 8 | Medium land area |
| 4 | $1.68 \times 10^6$–$3.16 \times 10^6$ | 6 | Medium–large land area |
| 5 | $>3.16 \times 10^6$ | 4 | Large land area |

| Population variation in 1990–2018 (P) | | | |
|---|---|---|---|
| **Grade** | **Population variation (P)** | **Number of countries** | **Description** |
| 1 | $<5.74 \times 10^5$ | 33 | Small population variation |
| 2 | $5.74 \times 10^5$–$1.90 \times 10^6$ | 15 | Small–medium population variation |
| 3 | $1.90 \times 10^6$–$4.95 \times 10^6$ | 7 | Medium population variation |
| 4 | $4.95 \times 10^6$–$1.050 \times 10^7$ | 3 | Medium–large population variation |
| 5 | $>1.050 \times 10^7$ | 2 | Large population variation |

| Urbanisation rate variationin 1990–2018 (U) | | | |
|---|---|---|---|
| **Grade** | **Urbanisation rate variation (U)** | **Number of countries** | **Description** |
| 1 | <2.06 | 10 | Small urbanisation rate variation |
| 2 | 2.06–8.48 | 23 | Small–medium urbanisation rate variation |
| 3 | 8.48–15.31 | 14 | Medium urbanisation rate variation |
| 4 | 15.31–22.96 | 10 | Medium–large urbanisation rate variation |
| 5 | >22.96 | 3 | Large urbanisation rate variation |

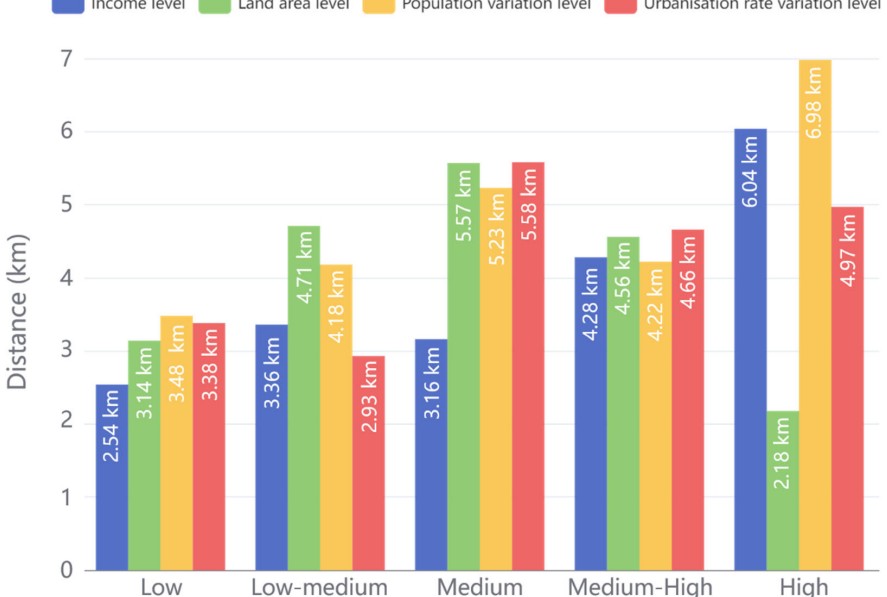

**Figure 18.** The average distance variation in relation to the indicators of the countries in which cultural heritage sites are located.

Thus, high levels of national income, medium land area, high population variation and rapid urbanisation are all influential factors in distance variations between world cultural heritage sites and neighbouring towns, with urbanisation in particular having a significant influence. Countries that possess these characteristics or a combination of them should pay attention to the harmonious relationship between urban development and cultural heritage in the future.

## 5. Discussion

In the course of this study, we also paid attention to other factors that influence distance variations between world cultural heritage sites and neighbouring towns; these factors include the effects of economic and political regionalisation and the specific conditions of different countries.

The economic systems and structures of the world today are undergoing great variations, and economic and political regionalisation and globalisation are the general trends. In particular, the transformation of cities in a region into urban agglomerations and city circles and the changing distances between cities and neighbouring heritage sites require more attention. First, with more frequent intercity interaction and the blurring of boundaries between cities, suburban areas adjacent to cities are the best spaces for urban expansion; therefore, the cultural heritage located in these locations will be the first to be affected by urban development. Second, once a certain scale of urban agglomeration has occurred in a number of cities, development will not stop but will continue to expand to the periphery at a faster pace. Cultural heritage sites located on the periphery of these cities are then affected by imminent urban expansion.

Different countries have very different natural and social conditions and have undergone very different historical development processes; consequently, there are significant differences in the number, location and type of cultural heritage sites that these countries possess. Additionally, some countries have a rich cultural heritage of their own, but for various reasons, they have not been able to nominate themselves as world cultural heritage sites. Additionally, some countries are at different stages of development; for example, China and India are at a stage of fast economic growth and have rapidly growing urbanisation rates and populations. Although the number of world cultural heritage sites is relatively small, most are affected to varying degrees by urban expansion. However, most European countries are strong, and although they have not experienced much growth in urbanisation rates or population, these countries have many world cultural heritage sites that are still greatly affected by urban expansion.

In addition, although urban expansion is a dominant trend in global development, some heritage sites, owing to better conservation measures in some countries, have remained near urban areas or are being preserved and are maintaining their value without being disturbed by urban expansion.

## 6. Conclusions

This article examines the spatial relationship between world cultural heritage and neighbouring towns and the evolution of this relationship. The results of this study show that over the past 28 years, 8.52% of world cultural heritage sites have been incorporated into neighbouring towns because of urban expansion and that the growth in the number of these sites can be divided into three phases, with two periods of rapid growth. The spatial distribution of these heritage sites is somewhat aggregated, mainly in Europe and the Asia-Pacific region, with a Gini coefficient of 0.6918. Of the world cultural heritage sites that have historically been located outside urban areas, 523 have not yet undergone substantial changes in their spatial relationship with urban areas. However, the distance between most of these sites and neighbouring towns, mainly in Europe and the Asia-Pacific region, has been decreasing, with the distance of 19% being reduced by more than 7.6 km. Further analysis of the distance variation between cultural heritage sites and neighbouring towns reflects a relationship between heritage and indicators, such as the

income level and urbanisation rate of the country to which a site belongs. In particular, the urbanisation rate of the country most greatly affects the distance between heritage sites and neighbouring towns, thereby suggesting that countries undergoing rapid urbanisation should pay particular attention to the impact of urban expansion on cultural heritage.

In this article, the analysis of the mechanisms influencing the distance variation between world cultural heritage sites and neighbouring towns is conducted from a macroscopic perspective. However, especially for policy and planning development, the macro scale is far from adequate. In the future, the research could be refined at a more local scale. If as many indicators as possible are applied to the city and district scales surrounding cultural heritage sites, we will be able to better understand the relevance and impact mechanisms of cultural heritage sites and urban areas.

Building on this article, heritage sites with large distance variations from neighbouring towns will be an important focus of our future research. We will use high-resolution remote sensing data to extract information at the individual site scale and monitor land use changes in the neighbouring towns. Based on the monitoring results, we will assess the impact of urban expansion and land-use change on heritage sites and provide timely reporting and early warning to heritage management authorities.

Relying on Big Earth Data, in this article, we monitor and assess the spatial relationship between cultural heritage sites and neighbouring towns at the global scale. Assessing these relationships according to the objectives of SDG 11.4 enables us to better assess the sustainability and status of cultural heritage. In the case that SDG11.4.1 indicators are not assessable by lack of data [40], the study has some reference value in particular. We believe cultural heritage sites that are becoming part of urban areas should be given adequate attention and protection, as these sites are of profound significance for the continuation and sustainable development of human civilisation.

**Author Contributions:** Conceptualisation: R.Y., Y.L. and Y.X.; data curation: Y.L., R.Y. and Y.X.; formal analysis: Y.X., Y.L. and R.Y.; methodology: Y.X., R.Y. and Y.L.; visualisation: Y.X. and R.Y.; writing—original draft preparation: Y.X. and R.Y.; writing—review and editing: Y.X., R.Y., W.L. and F.C.; funding acquisition: R.Y. All authors have read and agreed to the published version of the manuscript.

**Funding:** The study is funded by the Earth Big Data Science Project of the Chinese Academy of Sciences Pilot Project (No. XDA19030502, No. XDA19030501).

**Data Availability Statement:** The data for distance between heritages sites and neighbouring towns (1990–2018) which is the core of this study has been produced as a dataset and is available at: http://doi.org/10.57760/sciencedb.02340 for any individual or organisation that requires it.

**Acknowledgments:** The authors acknowledge all the data sources mentioned in the article for providing the open-source big data and are thankful for all the helpful comments provided by the reviewers and editors. We are also very grateful to Wang Shaohua, Zhu Guobin, Zhang Mu, Zhou Songtao, Han Ge, Wang Huamin form the School of Remote Sensing and Information Engineering, Wuhan University for providing professional suggestions for this study.

**Conflicts of Interest:** The authors declare no conflict of interest.

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
