# Peer review of "The Spatial Relationship and Evolution of World Cultural Heritage Sites and Neighbouring Towns"

_remotesensing, doi:10.3390/rs14194724_

Round 1

Reviewer 1 Report (Previous Reviewer 2)

The Authors provided an updated manuscript, but it does not address the Reviewer concerns outlined in the previous review round. Although the article is written on an important topic, two key problems remain.

QUOTE FROM PREVIOUS REVIEW

Problem 1. The article hardly fits the Journal Scope and the Scope of this special issue “Special Issue "Global Urban Observation for SDG Goal 11: Sustainable Cities and Communities": https://www.mdpi.com/journal/remotesensing/about

https://www.mdpi.com/journal/remotesensing/special_issues/urban_SDG

It does use a dataset that is derived from a remote sensing dataset "GUB dataset was developed by Prof. Gong Peng's team at Tsinghua University in 2020 based on the developed global high-resolution (30 m) artificial impervious surface mapping product (Global Artificial Impervious Area (GAIA))", but this dataset is used just as a reference to cities' extents.

The article does not contain research that is clearly within the Remote Sensing domain.

END OF QUOTE

The Authors claim that “Without remote sensing methods, this study would not exist.” This is not 100% true, as Authors use the remote sensing dataset just as a refence to cities' extents which could be derived from other sources to base the study on. A paper that is just using a remote sensing dataset does not automatically fit the Remote Sensing domain.

Moreover, in their response letter Authors claim that the study fits into “SDG11.4, "Strengthening efforts to protect and safeguard the world's cultural and natural heritage".”, but in Conclusions Authors write “The analysis in this paper … is conducted here from a macroscopic perspective. However, especially for policy and planning development, the macro scale is far from adequate.”. So, the Reviewer is wondering how the paper and its outcomes will strengthen the efforts to protect and safeguard the world's cultural and natural heritage (goal SDG11.4)?

QUOTE FROM PREVIOUS REVIEW

Problem 2. The methods used in the article are trivial and provide little new knowledge.

END OF QUOTE

Authors reply that “the conclusions are meaningful”. At the same time, Authors write in Conclusions that “The analysis in this paper … is conducted here from a macroscopic perspective. However, especially for policy and planning development, the macro scale is far from adequate.”. Hence, how the Conclusions can be treated as meaningful?

Next, the paper can be hardly original and meaningful only because it highlights the concern of merging World Cultural Heritage Sites into Neighbouring Towns by providing simple distance statistics. There are other papers that also address similar questions, e.g. https://www.tandfonline.com/doi/full/10.1080/20964471.2020.1853362

The contributions of the paper are quite thin for a Q1 journal.

Authors write “Based on Big Earth Data, in this paper, we carry out…”. What are the criteria for Big Data and what is the volume of the data the Authors use directly?

Conclusion. The paper must include more in-depth methods and more impactful results in the context of “Sustainable Cities and Communities". 

Author Response

Reviewer 2 Report (Previous Reviewer 3)

The Authors addressed all my comments, and expanded on the discussion of results (in the Results section) plus further specification of the methodology.

Arcgis (line 153) --> ArcGIS

Author Response

Response to Reviewer 2 comments

Thank you for all your comments concerning our manuscript entitled “The Spatial Relationship and Evolution of World Cultural Heritage Sites and Neighbouring Towns”. We have read through them carefully and have made corrections. There are also a few other revisions as follows.

  1. We provided a dataset of the article.
  2. We updated the description of the data source, background and value of the study, and some future plans.
  3. We added 5 new references (number 7, 16-18, and 37).
  4. We revised some details and corrected typographical errors.

Considering the reviewer's suggestion, we have special thanks to you for your helpful comments. We hope that we have now produced a more balanced and better account of our work so that it accords more with the requirements for publication.

This manuscript is a resubmission of an earlier submission. The following is a list of the peer review reports and author responses from that submission.

Round 1

Reviewer 1 Report

pl. see my attached comments

Reviewer 2 Report

The article is written on an important topic.

However, there are two key problems.

Problem 1. The article hardly fits the Journal Scope and the Scope of this special issue “Special Issue "Global Urban Observation for SDG Goal 11: Sustainable Cities and Communities": https://www.mdpi.com/journal/remotesensing/about

https://www.mdpi.com/journal/remotesensing/special_issues/urban_SDG

It does use a dataset that is derived from a remote sensing dataset "GUB dataset was developed by Prof. Gong Peng's team at Tsinghua University in 2020 based on the developed global high-resolution (30 m) artificial impervious surface mapping product (Global Artificial Impervious Area (GAIA))", but this dataset is used just as a reference to cities' extents.

The article does not contain research that is clearly within the Remote Sensing domain.

Problem 2. The methods used in the article are trivial and provide little new knowledge.

Conclusion

It would be natural to move this article to a more relevant journal, e.g., Sustainability, which articles are not necessarily from the Remote Sensing domain.

Reviewer 3 Report

This well-written paper is intended to study the spatial relationship between recognised world heritage sites and the urban areas. This research becomes relevant and necessary given the rapid urban growth occurring in the latest decades, and the need to protect heritage sites against identity loss, and destruction. The Authors analyse multiple relevant factors affecting this growth.

A few comments to improve the paper:

- The limitations of this study should be described in detail.

- Please include a paragraph at the end of the Introduction describing how the rest of the paper is organised.

- Figures and tables should be mentioned in the text before they are displayed.

- Figure 11 should have in its legend each colour assigned to each period.

- Lines 376-377: phrase "resulting in [...] neighbouring towns" should be removed since it appears at the beginning of the sentence.